# A Case of IgG and IgA Anti-Laminin-332 Antibody-Positive Mucous Membrane Pemphigoid with IgG and IgA Anti-Envoplakin and Anti-Periplakin Antibodies

**Yoshiaki Matsushima [1], Masako Kitano [2], Daisuke Hayashi [3], Hiroyuki Goto [1], Mako Mine [3], Takeshi Yokoe [4], Makoto Kondo [1], Koji Habe [1], Yuji Toiyama [4], Takashi Hashimoto [3], Daisuke Tsuruta [3], Kazuhiko Takeuchi [2] and Keiichi Yamanaka [1],***

1   Department of Dermatology, Graduate School of Medicine, Mie University, Tsu 514-8507, Japan
2   Department of Otorhinolaryngology—Head & Neck Surgery, Graduate School of Medicine, Mie University, Tsu 514-8507, Japan
3   Department of Dermatology, Graduate School of Medicine, Osaka Metropolitan University, Osaka 545-8585, Japan
4   Department of Gastrointestinal and Pediatric Surgery, Graduate School of Medicine, Mie University, Tsu 514-8507, Japan
*   Correspondence: yamake@med.mie-u.ac.jp; Tel.: +81-59-231-5025; Fax: +81-59-231-5206

**Abstract:** A 76-year-old Japanese man presented with a 6-year history of a sore throat. He was treated at several clinics without any improvement before being referred to us. Physical examination revealed widespread erosions and ulcers from the palate to the larynx. Approximately 25 × 15 mm in size, erosive lesions were present on the retroauricular regions, forearms, and glans penis. Pseudomembranous conjunctivitis was also observed. The skin biopsy revealed a partial cleft formation below the epidermis, suggesting subepidermal bullous disease. Immuno-serological tests were negative for anti-desmoglein 1 (Dsg1), anti-Dsg3, anti-BP180, and anti-BP230 antibodies by ELISAs. A whole-body examination revealed gastric cancer. The possibility of mucous membrane pemphigoid (MMP) or paraneoplastic pemphigus (PNP) was considered. Indirect immunofluorescence using rat bladders showed positive IgG reactivity with cell surfaces on the transitional epithelia. Immunoblotting using recombinant proteins of laminin-332 showed both IgG and IgA reactivities with laminin-$\alpha$3, and immunoblotting using normal human epidermal extract showed double-positive reactivities with envoplakin and periplakin for both IgG and IgA antibodies. Based on the clinical and histopathological features and results of various immuno-serological tests, our case was diagnosed as anti-laminin-332-type MMP with serological findings of PNP. Twenty days after laparoscopic gastrectomy, treatment with oral methylprednisolone 32 mg/day was initiated, and mucosal and skin lesions improved.

**Keywords:** mucous membrane pemphigoid; anti-laminin-332; laminin-$\alpha$3; envoplakin; periplakin

## 1. Introduction

Anti-laminin-332-type mucous membrane pemphigoid (MMP) accounts for 25% of all types of MMP, and is the second major type, following anti-BP180-type MMP [1–3]. In this manuscript, we report a case with refractory erosive lesions on the mucosae and the skin, which was finally diagnosed as an anti-laminin-332-type MMP with serological findings of paraneoplastic pemphigus (PNP).

## 2. Case Presentation

A 76-year-old Japanese man presented with a 6-year history of a sore throat. He was treated at several clinics without any improvement. He was referred to the otorhinolaryngology department and the dermatology department at Mie University Hospital because of the gradual worsening of his symptoms including dysphagia and skin lesions. Physical examination revealed widespread erosions and ulcers from the palate to the larynx (Figure 1a). Approximately 25 × 15 mm erosive lesions were present on the retroauricular regions (Figure 1b), forearms, and glans penis. These skin lesions occurred 2 months prior to our visit. Pseudomembranous conjunctivitis was also observed (Figure 1c). The symptoms of conjunctivitis occurred 9 months prior to our visit. The skin biopsy from the left forearm revealed a partial cleft formation beneath the epidermis (Figure 1d), suggesting subepidermal bullous disease. In addition, an oral mucosal biopsy revealed an infiltrate of eosinophils and neutrophils within the epidermis, with evidence of possible subepidermal blister formation. Immuno-serological tests were negative for anti-desmoglein 1 (Dsg1), anti-Dsg3, anti-BP180, and anti-BP230 antibodies by ELISAs, and for anti-keratinocyte cell surface and anti-basement membrane zone antibodies by indirect immunofluorescence. Although no definitive diagnosis of autoimmune bullous disease was made, oral prednisolone 20 mg/day was administered for 2 weeks, followed by 40 mg/day for another week. However, the mucosal lesions persisted. At this point, the therapeutic effect of prednisolone was judged to be insufficient, and it was tapered and terminated. Moreover, a whole-body examination revealed gastric cancer (Figure 1e).

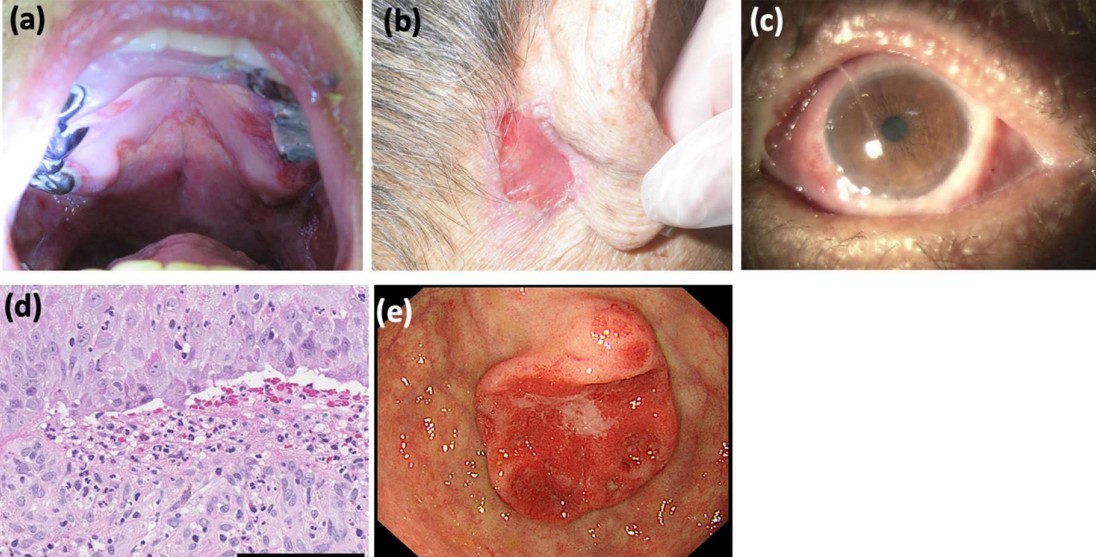

**Figure 1.** Clinical and histopathological findings. (**a**) The hard palate presented with erosion and ulcers. Erosions and ulcers were also present on the gingivae, the floor of the mouth, and from the pharynx to the esophagus. (**b**) An approximately 25 × 15 mm refractory erosive lesion observed on the retroauricular regions. (**c**) Pseudomembranous conjunctivitis and hyperemic conjunctiva. The upper eyelid adhered to the conjunctiva bulbi and could not be inverted. (**d**) Histopathology for the skin biopsy from left forearm showed subepidermal vesiculation and cleft formation at the dermoepidermal junction. In the uppermost dermis, there was an inflammatory cell infiltrate mainly of neutrophils with a few lymphocytes and eosinophils (hematoxylin-eosin staining; scale bar = 100 μm). (**e**) Gastroscopy detected a gastric cancer, 35 × 24 mm in size, which was histologically a well-differentiated adenocarcinoma.

One month after prednisolone discontinuation, the retroauricular erosive lesions recurred, and mucosal lesions on the throat worsened. Since the possibility of MMP or PNP was considered, further immuno-serological tests were performed [4]. Indirect

immunofluorescence using rat bladder showed positive IgG reactivity with cell surfaces of the transitional epithelia (Figure 2a). Immunoblotting using recombinant proteins of laminin-332 showed clear IgG and weak IgA reactivities with laminin-$\alpha$3 (Figure 2b), and immunoblotting using normal human epidermal extract showed double-positive reactivities with envoplakin and periplakin for both IgG and IgA antibodies (Figure 2c). The immunoblotting methods were previously described in detail [5]. In brief, the recombinant protein of laminin-332 (Oriental Yeast Co., Ltd., Tokyo, Japan) or normal human epidermal extract was directly solubilized in Laemmli's sample buffer, applied on 3–8% SDS-polyacrylamide gels, and transferred onto PVDF membrane. The strips of blotted membranes were blocked with Tris-buffered saline with 3% skim milk (dilution buffer) for 1 h at room temperature. The strips were incubated overnight at 4 °C with patients' sera diluted at 1:50 in dilution buffer. The strips were then incubated with horseradish peroxidase-conjugated polyclonal rabbit anti-human IgG antiserum (1:500–1:1000, gamma-chain-specific) (Dako, Glostrup, Denmark) as a secondary antibody for 1 h at room temperature. Finally, the reactions were detected by chemiluminescence using Super Signal West Dura Extended Duration Substrate for HRP (Thermo Fisher Scientific, Waltham, MA, USA).

Based on the clinical and histopathological features and results of various immuno-serological tests, our case was diagnosed as an anti-laminin-332-type MMP with serological findings of PNP. Twenty days after laparoscopic gastrectomy, the patient was treated with intravenous prednisolone 10 mg/day for 20 days, but the mucosal lesions did not resolve. Therefore, oral methylprednisolone 32 mg/day was started, and the mucosal and skin lesions improved. The methylprednisolone dose was gradually decreased to 12 mg/day without any recurrence of mucosal and cutaneous lesions.

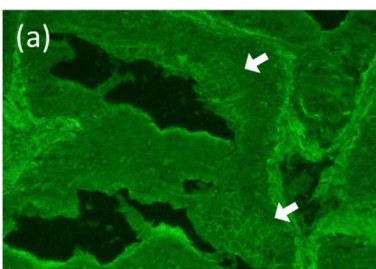 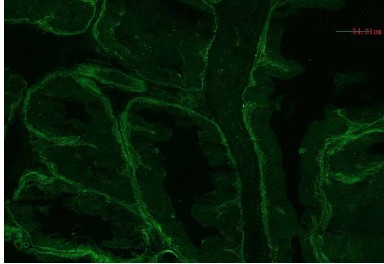 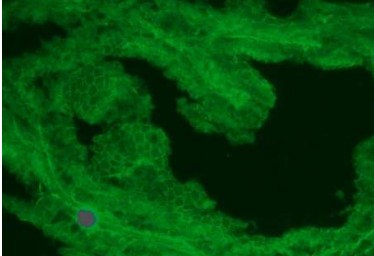

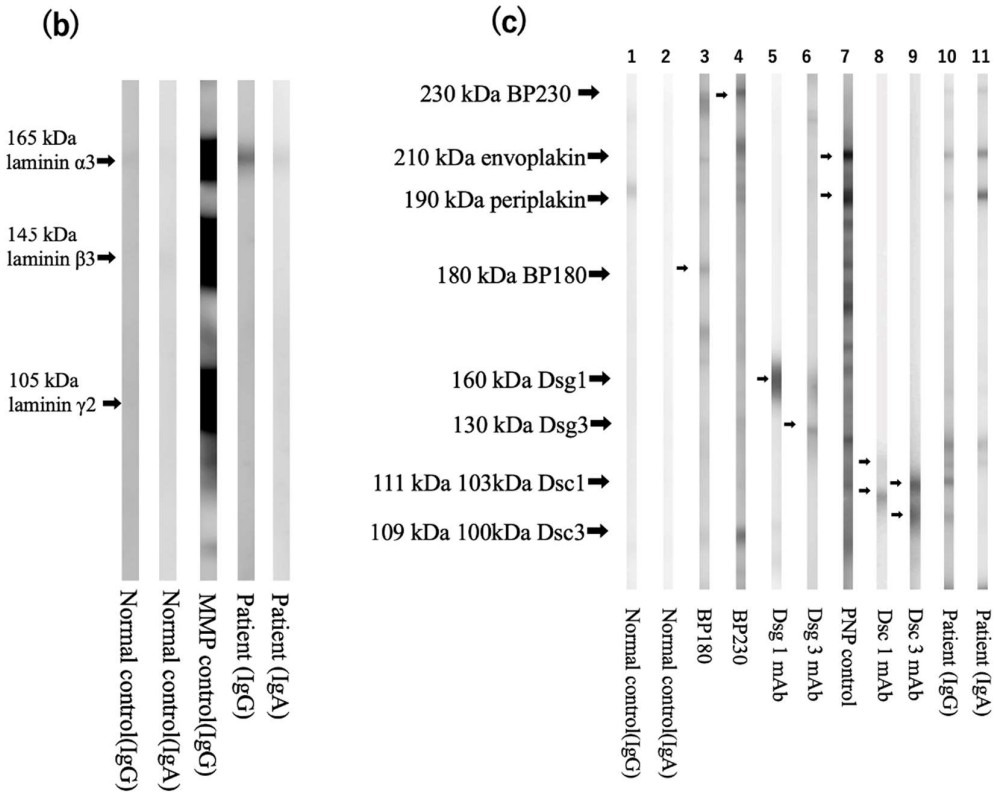

**Figure 2.** Immuno-serological findings (**a**) Left: indirect immunofluorescence using rat bladder showed IgG reactivity with cell surfaces of the transitional epithelia in this patient (white arrows, serum dilution ×40). Center: negative control. Right: positive control. (**b**) The results of immunoblotting using recombinant proteins of laminin-332. Normal control serum showed no specific reactivity for both IgG and IgA (lane 1, 2). IgG antibodies of control anti-laminin-332 MMP serum reacted with the 165 kDa laminin-$\alpha$3, 145 kDa laminin-β3, and 105 kDa laminin-γ2 (lane 3). IgG and IgA antibodies in our case's serum reacted with the 165 kDa laminin-$\alpha$3 (lane 4,5). (**c**) The results of immunoblotting using normal human epidermal extract. Normal control serum showed no specific reactivity (lanes 1,2). Control bullous pemphigoid (BP) sera reacted with the 180 kDa BP180 and 230 kDa BP230 (lanes 3,4). Specific monoclonal antibodies (mAbs) reacted with 160 kDa desmoglein 1 (Dsg1) (lane 5), the 130 kDa Dsg3 (lane 6), the 111 kDa and 103 kDa desmocollin 1 (Dsc1) (lane 8), and the 109 kDa and 100 kDa Dsc3 (lane 9). Control PNP serum reacted with the 210 kDa envoplakin and the 190 kDa periplakin (lane 7). IgG and IgA antibodies in our patient's serum reacted with envoplakin and periplakin (lanes 10,11). IgG antibodies in our case's serum also reacted with Dsc3 (lane 10).

## 3. Discussion

MMP is an autoimmune bullous disease that primarily affects the mucous membranes. Anti-laminin-332-type MMP patients show severer and more refractory lesions on multiple mucous membranes, including larynx and conjunctivae [1–3]. Anti-laminin-332-type MMP has been reported to be frequently associated with solid malignant tumors, particularly gastric cancer [1–3]. Our case showed the widespread mucosal lesions from throat to esophagus, resulting in dysphagia. In addition, he had gastric cancer.

Furthermore, our case showed double-positive reactivities to envoplakin and periplakin for both IgG and IgA antibodies, which are autoantibodies thought to be associated with PNP, as well as IgG reactivity with rat bladder epithelium [4,6]. Although the majority of PNP cases are associated with hematologic tumors, solid tumors are also occasionally found in PNP [6]. However, we finally diagnosed this case as MMP. This is

because the patient did not have the bloody crusts or ulcers on the lips that are commonly seen in PNP [7], his tumor was gastric cancer, which is specifically associated with anti-laminin-332-type MMP, and there was marked improvement of mucosal lesions with relatively low dose of systemic steroids after tumor resection.

Interestingly, our case showed both IgG and IgA antibodies to both laminin-$\alpha$3 and envoplakin/periplakin. The significance of these findings is currently unknown, but class-switch recombination in B-cells might be involved in the production of the immunoglobulins of multiple subclasses. The cause of the PNP-related autoantibody production is unknown in our case, but gastric cancer is not only associated with MMP, but might also potentially cause PNP. Alternatively, the production of multiple autoantibodies may be due to the epitope spreading phenomena.

## 4. Conclusions

Refractory mucosal erosions and ulcers may be autoimmune bullous disease and require histological and serological examination for diagnosis. Although anti-envoplakin and anti-periplakin antibodies are autoantibodies that are thought to be associated with PNP, their pathogenicity has not been clarified, and they may be also detected in other disease types such as MMP as in this case.

**Author Contributions:** Conceptualization, Y.M. and K.Y.; investigation, Y.M., D.H., M.M., T.H., D.T. and K.Y.; take care of the patient, Y.M., M.K. (Masako Kitano), H.G., T.Y., M.K. (Makoto Kondo), Y.T., K.H. and K.T.; writing—original draft preparation, Y.M.; writing—review and editing, T.H., D.T. and K.Y.; supervision, K.H.; project administration, K.Y. All authors have read and agreed to the published version of the manuscript.

**Funding:** This research received no external funding.

**Institutional Review Board Statement:** Ethical review and approval were waived for this study because of single case report.

**Informed Consent Statement:** Written informed consent has been obtained from the patient to publish this paper.

**Conflicts of Interest:** The authors declare no conflict of interest.

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
