# Peer review of "A Case of IgG and IgA Anti-Laminin-332 Antibody-Positive Mucous Membrane Pemphigoid with IgG and IgA Anti-Envoplakin and Anti-Periplakin Antibodies"

_dermatopathology, doi:10.3390/dermatopathology9030034_

Round 1
Reviewer 1 Report
The reviewer wishes to thank the editor and the authors for the opportunity to review this beautifully written, concise, well-illustrated, and thought provoking manuscript. This article is that of a case report of a patient with long standing likely mucosal erosions and ulcerations (“sore throat”) with subsequent serologies concerning for both anti-laminin 332-type MMP and PNP. It is of particular interest in that, given the physical exam and subsequent diagnosis of gastric cancer, a diagnosis of anti-laminin 332 MMP was rendered with serologies of PNP.
This is a serologically challenging case. While I agree that the clinical favors anti-laminin 332 MMP, it would be of interest/ perhaps augment the manuscript to further elucidate the time course of various symptoms. That is to say clarification as to if the patient always had cutaneous findings or if they initially started as only a “sore throat” and then in recent times progressed to include cutaneous findings (and perhaps even the pseudomembranous conjunctivitis). A change or modification in clinical findings over time (or even suddenly) could lend support to the idea of epitope spreading as an etiology for both clinical and serologic findings.1
It is also of note that a previous case by Yamada et al. also describes a case with immunoreactivity for envoplakin, periplakin and IgG antibodies reacting to the γ2 subunit of the laminin-332 molecule. However, in this instance the clinical favored PNP.2
The only additional thought is that a higher power image of Figure 2a may make the article more attractive.
Reference:
11. Didona D, Di Zenzo G. Humoral Epitope Spreading in Autoimmune Bullous Diseases. Front Immunol. 2018,9,779. Published 2018 Apr 17. doi:10.3389/fimmu.2018.00779
22. Yamada, H., Nobeyama, Y., Matsuo, K., Ishiji, T., Takeuchi, T., Fukuda, S., Hashimoto, T. and Nakagawa, H.. A case of paraneoplastic pemphigus associated with triple malignancies in combination with antilaminin-332 mucous membrane pemphigoid. Br. J. Dermatol. 2011, 166,1,230-231.
Author Response
Responses to the comments of Reviewer #1
Comments to the Author:
- While I agree that the clinical favors anti-laminin 332 MMP, it would be of interest/ perhaps augment the manuscript to further elucidate the time course of various symptoms. That is to say clarification as to if the patient always had cutaneous findings or if they initially started as only a “sore throat” and then in recent times progressed to include cutaneous findings (and perhaps even the pseudomembranous conjunctivitis).
Response: Thank you for your great suggestion. As you pointed out, the patient showed initially sore throat, which was followed by some other symptoms for several years. Ocular symptoms had appeared 9 months before he visited us, and skin symptoms had appeared 2 months before he visited us. We have supplemented the clinical course in the revised manuscript.
- The only additional thought is that a higher power image of Figure 2a may make the article more attractive.
Response: Thank you for your suggestion. High power image and pictures for negative and positive controls have been supplemented in Figure 2a.
Reviewer 2 Report
Was DIF performed to diagnose MMP? If not, how can the authors conclude that the patient suffers from MMP.
Since most conclusions come from Western blot testing, it is helpful to provide detailed information about the procedure and reagents used.
It cannot be concluded that the patient had anti-Periplakin IgG since there was reactivity at 190 kDa in control lane 1
IIF on rat bladder is not convincing at all, please provide illustrative image with negative and positive control.
Was indirect immunofluorescence on LN332 transfected cells performed?
What is the source of recombinant LN332?
Author Response
Responses to the comments of Reviewer #2
Comments to the Author:
- Was DIF performed to diagnose MMP? If not, how can the authors conclude that the patient suffers from MMP.
Response: Thank you for your suggestion. DIF was not performed in this patient. However, the diagnosis of MMP could be made clinically by refractory mucosal ulcers and pseudomembranous conjunctivitis, histopathologically by subepithelial blisters on both skin and oral mucosal biopsies, and serologically by anti-laminin 332 antibodies.
- Since most conclusions come from Western blot testing, it is helpful to provide detailed information about the procedure and reagents used.
Response: Thank you for your great suggestion. We have supplemented the description of Western blotting procedures and reagents.
- It cannot be concluded that the patient had anti-Periplakin IgG since there was reactivity at 190 kDa in control lane 1
Response: Thank you for your suggestion. The reaction in lane 1 you pointed out is non-specific reactivity with unknown protein, which is a litter larger than the 190 kDa periplakin.
- IIF on rat bladder is not convincing at all, please provide illustrative image with negative and positive control.
Response: According to this suggestion, we have added images for negative and positive controls in the revised manuscript. We appreciated for your suggestion.
- Was indirect immunofluorescence on LN332 transfected cells performed?
Response: Thank you for your suggestion. We did not perform indirect immunofluorescence on LN332 transfected cells. Instead, we routinely perform Western blotting using LM332 recombinant proteins for the detection of antibodies to LM332 subunits.
- What is the source of recombinant LN332?
Response: Thank you for your suggestion. Recombinant Human Laminin-5 (Oriental Yeast Co., Ltd., Tokyo, Japan) was used as the recombinant protein for laminin-332.
Reviewer 3 Report
This is an interesting case report of a patient with MMP that showed serologic features of PNP as well. My comments/questions are below.
Abstract, Case presentation and figure 1 legend: Change “Fingerhead-sized” to a scientific measurement.
Figure 1 legend: I would not describe the histopathologic findings as showing “liquefaction degeneration”. I would simple state there is subepidermal vesiculation. There appears to be more eosinophils than neutrophils in the image.
Was a DIF biopsy not done?
Author Response
Responses to the comments of Reviewer #3
Comments to the Author:
- Abstract, Case presentation and figure 1 legend: Change “Fingerhead-sized” to a scientific measurement.
Response: Thank you for your suggestion. We have changed the size to be expressed numerically.
- Figure 1 legend: I would not describe the histopathologic findings as showing “liquefaction degeneration”. I would simple state there is subepidermal vesiculation. There appears to be more eosinophils than neutrophils in the image.
Response: Thank you for your suggestion. We have changed “liquefaction degeneration” to “subepidermal vesiculation”. In this specimen, eosinophils are present, but neutrophils predominated.
- Was a DIF biopsy not done?
Response: Thank you for your suggestion.
We did not perform indirect immunofluorescence on LN332 transfected cells. Instead, we routinely perform immunoblotting using LM332 recombinant proteins for the detection of antibodies to LM332 subunits.